# Characteristic Analysis and Error Compensation Method of Space Vector Pulse Width Modulation-Based Driver for Permanent Magnet Synchronous Motors

**DOI:** 10.3390/s24247945

**Published:** 2024-12-12

**Authors:** Qihang Chen, Wanzhen Wu, Qianen He

**Affiliations:** 1School of Physics and Information Engineering, Fuzhou University, Fuzhou 350108, China; 221120006@fzu.edu.cn; 2School of Optoelectronic Science and Engineering, University of Electronic Science and Technology, Chengdu 611731, China; wuwz@std.uestc.edu.cn

**Keywords:** driving error compensation, permanent magnet synchronous motor, space vector pulse width modulation, speed control

## Abstract

Permanent magnet synchronous motors (PMSMs) are widely used in a variety of fields such as aviation, aerospace, marine, and industry due to their high angular position accuracy, energy conversion efficiency, and fast response. However, driving errors caused by the non-ideal characteristics of the driver negatively affect motor control accuracy. Compensating for the errors arising from the non-ideal characteristics of the driver demonstrates substantial practical value in enhancing control accuracy, improving dynamic performance, minimizing vibration and noise, optimizing energy efficiency, and bolstering system robustness. To address this, the mechanism behind these non-ideal characteristics is analyzed based on the principles of space vector pulse width modulation (SVPWM) and its circuit structure. Tests are then conducted to examine the actual driver characteristics and verify the analysis. Building on this, a real-time compensation method is proposed, physically matched to the driver. Using the volt–second equivalence principle, an input–output voltage model of the driver is derived, with model parameters estimated from test data. The driving error is then compensated with a voltage method based on the model. The results of simulations and experiments show that the proposed method effectively mitigates the influence of the driver’s non-ideal characteristics, improving the driving and speed control accuracies by 88.07% (reducing the voltage error from 0.7345 V to 0.0879 V for a drastic command voltage with a sinusoidal amplitude of 10 V and a frequency of 50 Hz) and 53.08% (reducing the speed error from 0.0130°/s to 0.0061°/s for a lower command speed with a sinusoidal amplitude of 20° and a frequency of 0.1 Hz), respectively, in terms of the root mean square errors. This method is cost-effective, practical, and significantly enhances the control performance of PMSMs.

## 1. Introduction

Permanent magnet synchronous motors (PMSMs) are widely used across various fields, including aviation, aerospace, marine, and industry, due to their high angular position accuracy, energy conversion efficiency, and fast response. Figure 1 illustrates the classical drive–control framework for a PMSM, serving as an example of a three-phase motor digital drive–control system. The framework primarily consists of a controller, a driver, and a PMSM. Command/reference voltages are sent from the controller to the driver as pulse width modulation (PWM) waves (three pairs/six channels). The driver then generates AC voltage to drive the PMSM using DC power according to the PWM signal. Phase currents and rotor angles are sampled by sensors and fed back to the controller, which calculates the command voltage for the next control cycle based on the feedback and reference.

In a PMSM drive–control system, the driver includes various power electronic components, with the insulated gate bipolar transistor (IGBT) as the core, functioning as a switch. The ideal step characteristics of IGBT on–off switching are not achievable due to material limitations, manufacturing processes, and environmental factors. Instead, a transition process and steady-state error typically occur, known as the driver’s non-ideal characteristics. Additionally, to prevent short-circuiting the DC power when switching IGBT states, as shown in Figure 1, a delay between switching states, called the driver’s dead time [1], is required. These non-ideal characteristics and the dead time cause deviations between the actual and reference driving voltages, leading to harmonic voltages [2,3], current distortion [4], instability in torque and speed, and increased vibrations and noise [5,6,7], negatively affecting PMSM stability and reliability [8]. Therefore, addressing the non-ideal characteristics of the driver is critical to improving motor control performance.

Compensation methods for the non-ideal characteristics of the drive fall into two categories: hardware and software. Early research focused on hardware solutions to address the discrepancies in drive output voltage. For example, Ref. [9] compensated for voltage differences by adding a voltage measurement module. However, as demand for cost-effectiveness and flexibility increased, research has shifted toward software-based solutions. Ref. [3] established a voltage observer based on the mathematical model of a permanent magnet synchronous motor, using error feedback for online compensation. Other studies developed software compensation through the ideal dead zone model of the IGBT. For example, Ref. [10] analyzed switching errors via this model; however, such methods rely heavily on accurate model parameters, and the ideal model often deviates significantly from actual conditions. Refs. [11,12] addressed this by developing an output voltage model of the driver, considering the deadband, but these models still overlook the transition process and other factors that affect IGBT switching, limiting their accuracy.

Recent research has focused on modern control algorithms and neural networks for compensating non-ideal drive characteristics. Ref. [13] applied finite control set model predictive control (FCS-MPC) and digital sliding mode control (DSMC) to estimate and compensate for input–output errors caused by deadband effects. Ref. [14] used neural networks to identify driver nonlinearity and decouple deadband and resistance voltages for online compensation of errors due to dead time effects. Ref. [15] proposed an LMS adaptive trap filter for online compensation, while Ref. [16] proposed a nonlinear approach to estimate and correct non-ideal characteristics under varying conditions. However, these methods often involve parameter adaptation processes and significantly increased computational complexity, hindering their ability to provide real-time control for PMSMs.

In contrast, this paper presents an error compensation method for non-ideal driver characteristics with the following contributions: (1) an analysis of the generation mechanisms of driver non-idealities based on the space vector pulse width modulation (SVPWM) principle and circuit structure and (2) test experiments designed to identify the primary factors influencing driver characteristics, leading to the development of a new real-time compensation method. Based on the volt–second equivalence principle, a driver input–output voltage model is theoretically derived, and test data are used to estimate model parameters. A voltage compensation method is applied to compensate for drive errors. Simulations and experiments show that the proposed method effectively mitigates the influence of driver non-idealities, improving driving accuracy by 88.07% (RMSE) and speed control accuracy by 53.08% (RMSE). This provides a simple and effective solution to minimize the influence of driver non-idealities on motor control performance.

The paper is organized as follows: Section 1 focuses on the SVPWM principle and the mechanisms behind driver non-idealities. Section 2 details the experimental design to identify factors affecting driver characteristics and presents the input–output model. Section 3 introduces the voltage compensation approach. Section 4 and Section 5 validate the method’s feasibility and effectiveness through simulations and experiments. Section 6 concludes with this study’s findings.

## 2. Characterization of Space Vector Pulse Width Modulated Drivers

### 2.1. Overview of SVPWM Techniques

SVPWM is an optimal PWM scheme as it processes the complex reference voltage vector $u_{s}^{*}$ as a whole, rather than modulating each of the three phases separately. This method improves the utilization of drive DC voltage, reduces harmonic losses, and minimizes torque pulsations. It also allows for the real-time generation of high-precision waveforms using a high-speed digital signal processor (DSP), making it widely adopted in high-performance motor speed control systems.

Figure 2a shows the equivalent circuit of a three-phase space vector modulation driver in a PMSM. It comprises three pairs of switches, designated $S_{1}-S_{4}$,
$S_{2}-S_{5}$, and $S_{3}-S_{6}$, with the constraint that at most, only one of each pair of switches can be in an active state at any given moment. Accordingly, the eight possible combinations of voltage vectors (S0, S1, …, S7) consist of six equal-mode non-zero vectors (
$u_{1}$, $u_{2}$, …, $u_{6}$) and two zero vectors ($u_{0}$ and $u_{7}$), depending on the switch state. These vectors represent the three-phase voltages when the motor phases are not floating. The endpoints of the six non-zero vectors form the vertices of a hexagon, as shown in Figure 2b, dividing the plane into six sectors (I, II, III, IV, V, VI).

When the upper switch is on, the two zero vectors $u_{0}$ and $u_{7}$ indicate that either all the lower or upper switches of the half-bridges in the driver are active. At this point, the amplitude of the voltage vector is zero, as the three-phase windings are short circuited.

To obtain the reference voltage vector $u_{s}^{*}$, it can be expressed as a linear combination of $u_{1}$, $u_{2}$, …, $u_{6}$. Based on the projection relation of the vectors and the volt–second equivalence principle, $u_{s}^{*}$ can be expressed as follows [17]:(1)$$u_{s}^{*}=\frac{T_{n}}{T}u_{n}+\frac{T_{n+1}}{T}u_{n+1}.$$
where $T=T_{n}+T_{n+1}+T_{0}$, $T_{n}$, $T_{n+1}$, and $T_{0}$ are the durations of $u_{n}$, $u_{n+1}$, and $u_{0}$, respectively, and $u_{n}$ and $u_{n+1}$ (n=1,2,...,6) are two boundary vectors of the sector where $u_{s}^{*}$ is located.

In conclusion, a specific $u_{s}^{*}$ can be achieved by regulating the combinations of states of the aforementioned three pairs of switches and their respective durations. The following section will analyze the switching process of the switch states.

To illustrate, consider the state-switching of $S_{1}-S_{4}$ in Figure 2a. If the corresponding switches are activated when the IGBT input is high (H) and deactivated when it is low (L), the ideal control signals for these pairs of switches are depicted in Figure 3a. The actual on–off time of the IGBTs must adhere to a specific period, denoted as $T_{on}$ and $T_{off}$. Consequently, if one of the switches is activated before the other is deactivated, the DC voltage source will be shorted. To prevent this scenario, it is essential to introduce a delay period to ensure the other switch is properly opened. This delay is referred to as the dead time, denoted as $T_{d}$. At this juncture, the control signal takes the form illustrated in Figure 3b, which is representative of the $S_{1}-S_{4}$ connection point (a phase terminal) potential $V_{a}$, as shown in Figure 3c. The actual change in drive voltage is not an ideal step-type transition but rather a complex process, which may lead the actual generated voltage vector to deviate from the prescribed value $u_{s}^{*}$, thereby generating a driving error.

### 2.2. IGBT Internal Structure and On–Off Time Analysis

An IGBT is a composite, fully controlled, voltage-driven power semiconductor device comprising a BJT (bipolar junction transistor) and a MOS (metal oxide semiconductor). The fifth-generation IGBT is employed in the driver, with its internal structure displayed in Figure 4a.

In Figure 4a, the N-emitting region is referred to as the source region, and the electrode attached to it is designated as the source (i.e., emitter E). The control region of the device is the gate region, and the electrode associated with it is called the gate (i.e., gate G). The channel is formed at the boundary immediately adjacent to the gate region. The section of the device comprising the P-type material between the collector (C) and emitter (E) is referred to as the subchannel region. The N base is known as the drain region. The opposite side of the leakage region of the P+ region is designated as the leakage injection region (drain injector). This is a distinctive functional area of the IGBT, formed by the conjunction of the leakage region and subchannel region, thereby constituting a PNP bipolar transistor. This configuration serves as the emitter region, wherein the leakage region injects holes, and conductive modulation is utilized to reduce the device’s on-state voltage. The electrode connected to the drain injection region is referred to as the drain (i.e., collector C) [18].

The ideal equivalent circuit of an IGBT module consists of a PNP bipolar transistor and a power MOSFET forming a Darlington connection to create a monolithic Bi-MOS transistor. A freewheeling diode (FWD), also known as a flywheel diode, is positioned between the CE collectors. The drain of the MOSFET is connected to the base of the PNP transistor, whereas the source of the MOSFET is connected to the collector of the PNP transistor, which serves as the emitter of the IGBT, as illustrated in Figure 4b.

Three parasitic capacitances are present within the IGBT: the gate–source capacitance ($C_{gs}$), the gate–drain capacitance ($C_{gd}$), and the drain–source capacitance ($C_{ds}$). Additionally, $L_{g}$, $L_{d}$, and $L_{s}$ represent the stray inductances of the gate, drain, and source, respectively, due to the device package. $C_{gdj}$ represents the gate cross-stack depletion layer capacitance, while $C_{bcj}$ denotes the internal parasitic capacitance of a PNP transistor. These two parameters are dependent on the applied voltage, which can be expressed as follows [19]:(2)$$C_{\mathrm{gdj}}\left(t\right)=\frac{A_{gd}\varepsilon_{si}}{\sqrt{\frac{2\varepsilon_{si}\left(V_{ds}\left(t\right)-V_{gs}\left(t\right)+V_{Td}\right)}{qN_{L}}}},$$
(3)$$C_{\mathrm{bcj}}\left(t\right)=\frac{A\varepsilon_{si}}{\sqrt{\frac{2\varepsilon_{si}V_{ds}\left(t\right)}{qN_{L}}}},$$
(4)$$C_{GE}\left(t\right)=C_{gs}+\frac{1}{\frac{1}{C_{gd}}+\frac{1}{C_{gdj}\left(t\right)}+\frac{1}{C_{bcj}\left(t\right)+C_{ds}}},$$
(5)$$C_{GD}\left(t\right)=\left\{\begin{array}{c} C_{gd},\\ \frac{C_{gd}C_{gdj}\left(t\right)}{C_{gd}+C_{gdj}\left(t\right)}, \end{array}\right.\begin{array}{c} V_{ds}\leq V_{GS}-V_{GE\left(th\right)},\\ V_{ds}\geq V_{GS}-V_{GE\left(th\right)}. \end{array}$$

The location of the aforementioned item is as follows: $C_{GE}\left(t\right)$ represents the total capacitance between the gate and emitter of the IGBT, while $C_{GD}\left(t\right)$ denotes the total capacitance between the gate and drain of the IGBT. Additionally, $\varepsilon_{si}$ signifies the silicon dielectric constant, whereas q is the unit electron charge. $V_{Td}$, on the other hand, is the gate–drain overlap depletion threshold voltage, which is typically approximated to 0. $V_{GE\left(th\right)}$ represents the threshold voltage for the conduction of the IGBT. $V_{GS}$ denotes the gate–source inter-electrode voltage of the internal MOS structure, while $V_{ds}$ signifies the drain–source inter-electrode voltage of the internal MOS structure. A is the effective working area of the chip, and $A_{gd}$ is the area of the gate and drain overlap. Finally, $N_{L}$ represents the doping concentration of the N base region. In the context of a switching transient, the drain–source voltage ($V_{ds}$) is approximately equal to the gate–source voltage ($V_{CE}$).

The IGBT on-time $T_{on}$ can be divided into the sum of the time for the gate voltage $V_{GE}$ to rise from 0 to the gate on-voltage $V_{GE\left(th\right)}$ and the time for the collector’s current $I_{C}$ to rise to $0.9I_{C}$, which increases rapidly after $V_{GE}$ exceeds $V_{GE\left(th\right)}$ and is much smaller with a charging time of $C_{GE}$. Before $I_{C}$ is generated, the drive current charges only $C_{GE}$, and the $V_{GE}$ rise curve can be expressed as
(6)$$V_{GE}\left(t\right)=V_{G\left(on\right)}\left(1-e^{-\frac{t}{R_{on}C_{GE}}}\right)$$
where $R_{on}$ is the gate turn-on resistance.

The time of this phase is expressed as
(7)$$t_{V_{GE\left(th\right)}}=-R_{on}C_{GE}\ln\left(1-\frac{V_{GE\left(th\right)}}{V_{G\left(on\right)}}\right).$$

As illustrated in Figure 2a, in the ideal scenario where both switches are in a closed position, the potential difference between the collector and emitter of each switch is $\frac{1}{2}U_{dc}$. During the charging process of $C_{GE}$, the equivalent capacitance between the collector and emitter poles, $C_{CE}$, also undergoes a charging process, resulting in a voltage between the collector and emitter, $V_{CE}$, of
(8)$$V_{CE}\left(t\right)=\frac{1}{2}U_{dc}\left(1-e^{-\frac{t}{R_{on}C_{CE}}}\right).$$

Once the gate voltage reaches the gate conduction voltage, the collector current ($I_{C}$) increases to $0.9I_{C}$. During this process, the driving current charges $C_{GC}$, which is equivalent to $C_{gd}$ in series with $C_{gdj}\left(t\right)$. The charging of the stray inductors, $L_{d}$ and $L_{s}$, significantly influences the value of $V_{CE}$ during this process. Therefore, $V_{CE}$ can be represented as
(9)$$\begin{array}{c} V_{CE}=\frac{1}{2}U_{dc}\left(1-e^{-\frac{t}{R_{on}C_{CE}}}\right)+\left(\frac{1}{2}U_{dc}-V_{CE}\left(t_{V_{GE\left(th\right)}}\right)\right)\cdot e^{-\left(t-t_{V_{GE\left(th\right)}}\right)\cdot \frac{L}{R_{on}}} \end{array}.$$
where L is the equivalent inductor of the stray inductors ($L_{d}$ and $L_{s}$) and $\left(\frac{1}{2}U_{dc}-V_{CE}\left(t_{V_{GE\left(th\right)}}\right)\right)$ is the voltage across the equivalent inductor L.

As the time taken for this process is much shorter than that of the previous one, the IGBT turn-on delay can be calculated as follows:(10)$$T_{on}=t_{V_{GE\left(th\right)}}+t_{0.9I_{CE}}\approx -R_{on}C_{GE}\ln\left(1-\frac{V_{GE\left(th\right)}}{V_{G\left(on\right)}}\right).$$

The IGBT turn-off time ($T_{off}$) is primarily a gate discharge process, which is largely influenced by the MOSFET structure. It can be decomposed into two components: the time required for the gate voltage to drop from $V_{GE\left(on\right)}$ to the Miller plateau voltage and the time required for the Miller plateau to be reached. This can be expressed as follows [20]:(11)$$T_{off}=t_{V_{G\left(on\right)}\to V_{Miller}}+t_{Miller}\;=R_{off}\left(C_{GS}+C_{GD}\right)ln\frac{g_{fs}V_{GE}}{g_{fs}V_{GE\left(th\right)}+I_{ds\left(max\right)}}+\frac{\left(V_{DM}-V_{on}\right)g_{fs}R_{off}C_{GD}}{I_{ds\left(max\right)}+g_{fs}V_{GE\left(th\right)}}.$$

The location of the aforementioned item is as follows: $R_{off}$ represents the gate shut-down resistance, $g_{fs}$ denotes the gate–source transconductance, $I_{ds\left(max\right)}$ signifies the maximum value of the channel current, $V_{DM}$ represents the maximum value of the drain voltage, and $V_{on}$ refers to the IGBT in the MOS structure of the on-state voltage drop.

The turn-off process for the gate voltage drops to $V_{GE\left(on\right)}$ before the $C_{CE}$ capacitor discharge and into the Miller platform after the stray inductor and the continuity diode to form a loop discharge. This constitutes two processes, in which $V_{CE}$ can be expressed as follows:(12)$$V_{CE}\left(t\right)=\left\{\begin{array}{cc} \frac{1}{2}U_{dc}\left(-e^{-\frac{t}{R_{on}C_{CE}}}\right) & when\;t<t_{m}\\ \begin{array}{c} \frac{1}{2}U_{dc}\cdot \left(-e^{-\frac{t}{R_{on}C_{CE}}}\right)\\ -V_{CE}(t_{m})\cdot e^{-(t-t_{m})\cdot \frac{L}{R_{on}}} \end{array} & when\;t>t_{m} \end{array}\right.$$
where $t_{m}$ represents the instance when entering the Miller plateau.

### 2.3. Effect of Voltage on IGBT Switching Time

When the IGBT is turned on, as the bus voltage increases, $V_{CE}$ (gate voltage) and $V_{ds}$ (drain–source voltage) increase. From Equations (2)–(4), both $C_{gdj}$ and $C_{bcj}$ decrease, causing $C_{GE}$ to decrease, leading to a decrease in $T_{on}$.

During the IGBT turn-off process, electron–hole pairs are generated in the depletion region of the $J_{2}$ junction. However, owing to the absence of electron and hole concentrations in the barrier area, recombination centers produce these carriers, which subsequently attempt to re-establish thermal equilibrium. As a result, they are expelled into the P-region and N-drift region (the base) as soon as new electron–hole pairs are generated.

As $V_{CE}$ increases, the width of the barrier area also increases, leading to the generation of more electron–hole pairs, which causes an increase in their concentration in the P- and N-drift regions. According to the theory of charge generation, a longer duration of the electron–hole complex results in an extended trail current time, thus prolonging the IGBT turn-off time as $V_{CE}$ increases. The derivation process follows:

Following the principle of charge control:(13)$$I_{c\left(BJT\right)}\left(t\right)=\frac{Q_{p}\left(t\right)}{T_{tp}\left(t\right)}$$
(14)$$T_{tp}\left(t\right)=\frac{{\left[W_{B}-W\left(t\right)\right]}^{2}}{4K_{A}D_{P}}$$
(15)$$K_{A}=A_{C}=A_{E}$$
where $Q_{p}\left(t\right)$ represents the hole charge in the N region to be compounded, $T_{tp}\left(t\right)$ represents the base hole transit time, $W_{B}$ represents the base width, $W\left(t\right)$ represents the depletion region width, $A_{C}$ represents the collector-region area of the PNP, $A_{E}$ represents the emitter-region area, and $D_{P}$ represents the base-region hole diffusion coefficient.

In the case of IGBT conduction, the width of the depletion region is zero, and it can be concluded that
(16)$$I_{0}=I_{MOS}+\frac{4K_{A}D_{P}Q_{P0}}{{W_{B}}^{2}}.$$
where $Q_{P0}$ is the base hole charge during conduction.

After the IGBT is turned off, the conductive channel disappears rapidly, $I_{MOS}=0$, and the current is as follows:(17)$$I_{1}=\frac{Q_{P0}}{\frac{{\left(W_{B}-W\right)}^{2}}{4K_{A}D_{P}}}=\frac{I_{0}-I_{MOS}}{{\left(1-\frac{W}{W_{B}}\right)}^{2}}.$$

In the BJT:(18)$$I_{C}=\beta I_{B}=\beta I_{MOS}.$$

Combining Equations (16)–(18) yields:(19)$$\begin{array}{cc} \Delta I & =I_{0}-I_{1}\\ & =I_{MOS}+I_{C\left(BJT\right)}-I_{C\left(BJT\right)}{\left(1-\frac{W}{W_{B}}\right)}^{-2}\\ & =I_{MOS}\left\{1-\beta \left[{\left(1-\frac{W}{W_{B}}\right)}^{-2}-1\right]\right\} \end{array}.$$

The width of the depletion region is [21]:(20)$$W={\left[\frac{2\varepsilon_{S}\left(V_{bi}+V_{R}\right)\left(N_{A}+N_{D}\right)}{eN_{A}N_{D}}\right]}^{\frac{1}{2}}.$$
where $V_{R}$ represents the reverse bias voltage in the depletion region, $\varepsilon_{S}$ represents the semiconductor dielectric constant, $V_{bi}$ represents the built-in potential difference at thermal equilibrium, $N_{A}$ represents the atomic density of the acceptor impurity, and $N_{D}$ represents the atomic density of the donor impurity.

In conclusion, W is proportional to $V_{R}$, which is proportional to $V_{CE}$. ΔI is inversely proportional to W. Consequently, as $V_{CE}$ increases, ΔI decreases. If the current remains constant, $I_{1}$ increases, resulting in a prolonged turn-off time.

Consequently, when the current remains constant, an increase in the bus voltage results in a reduction in the on-time and an extension of the off-time.

### 2.4. Effect of Current on IGBT Switching Time

When the IGBT is turned on, from Equations (2)–(4), the current has no effect on $C_{gs}$ when the voltage is constant. Consequently, it will not influence $T_{on}$.

When the IGBT first starts to turn off, the width of the depletion region is nearly zero, and the majority carriers complex current to the total current is:(21)$$K=\frac{\Delta I}{I_{0}}=\frac{I_{MOS}}{I_{MOS}+I_{C\left(BJT\right)}}=\frac{I_{MOS}}{I_{MOS}+\beta I_{MOS}}=\frac{1}{1+\beta }.$$

The following relationship exists for current gain in bipolar transistors [19]:(22)$$\alpha =\frac{\partial I_{C}}{\partial I_{E}}=\frac{I_{Cn}}{I_{En}+I_{R}+I_{EP}}=\gamma \cdot \alpha_{T}\cdot \delta .$$
(23)$$\beta =\frac{\partial I_{C}}{\partial I_{B}}=\frac{\alpha }{1-\alpha }.$$

In Equation (22), α represents the common base current gain, β represents the emitter current gain, γ represents the emitter injection efficiency, $\alpha_{T}$ represents the base region transport factor, and δ represents the complexation factor. When the collector current increases, the concentration of majority carriers in the base region increases, resulting in a decrease in γ. Owing to the surface complexation effect, the minority carriers also complex on the surface of the base region, which reduces $\alpha_{T}$ and thus leads to a decrease in α. Thus, α is inversely proportional to the current $I_{C}$, whereas from Equation (23), β is proportional to α, so β decreases when the current increases.

Based on the above analysis, the following conclusions are drawn:(24)$$K=\frac{1}{1+\beta }\propto 1-\frac{1}{I_{C}}.$$
(25)$$\frac{dK}{dI_{C}}\propto \frac{1}{{I_{C}}^{2}}.$$

If $dI_{C}$ is a constant value, dK decreases as $I_{C}$ increases and K decreases. For the same bus voltage, K decreases as the trail current increases, and the turn-off time increases. In summary, the smaller the current is, the longer the turn-off time for the same bus voltage.

## 3. Drive Characterization Modeling

### 3.1. IGBT On–Off Test Method

Inside the IPM, there are three bridges, U, V, and W, each composed of two IGBTs, positioned one above the other. To measure the switching characteristics of an IGBT, the remaining electronic switches are disconnected. In this paper, the tested IPM is the PM25RLA120. We specifically examined the U-phase lower half-bridge electronic switch, as illustrated in Figure 5.

The IGBT switching time under high and low voltages, as well as large and small currents, was tested in this experiment.

### 3.2. IGBT Conduction Test Analysis

From Figure 6a, when the DSP input control signal B[i] decreases for about 3 μs, $U_{R}$ begins to attenuate the oscillation increase, which is caused mainly by the optical coupling delay in the transmission process. In engineering applications, $U_{R}$ is considered to be on when it reaches 90% of the steady state. The dashed line in the subplot of Figure 6a illustrates the steady-state voltage at different bus voltages. With R= 10Ω and different $U_{dc}$ values, the switch-on time generally shows a decreasing trend with increasing $U_{dc}$, which is consistent with the analysis in Section 2.3.

From Figure 6b, at a constant voltage, the resistor R is changed. The switch-on time is almost constant, which is independent of the current, which is consistent with the analysis presented in Section 2.4.

### 3.3. Analysis of IGBT Turn-Off Testing

From Figure 7a, the DSP control signal increases for approximately 1.5 μs, $U_{R}$ begins to decrease, and $U_{R}$ decreases for approximately 6 μs to reach its minimum value. When the resistance is unchanged, the voltage increases and the current increases. As the current increases, the reverse current increases, and the time to reach the steady state (switch-off time) tends to increase for a certain load with increasing $U_{dc}$, which is consistent with the analysis presented in Section 2.4.

Figure 7b clearly shows that the turn-off time is significantly influenced by the tail current, with disconnection times ranging from approximately 115 to 140 μs, which is considerably longer than the turn-off time observed at higher currents.

Consequently, when considering Figure 7, it becomes apparent that the rates of voltage drop and tailing time for the IGBT are predominantly affected by the bus voltage. A higher bus voltage correlates with a faster turn-off speed; however, an increased tail current can substantially prolong the switch-off time. In practical applications, where the motor resistance is relatively low (the internal resistance of the torque motor we used is approximately 9.9 Ω), the control process primarily operates under high-current conditions; thus, this analysis focuses on scenarios involving high current.

### 3.4. Driver On–Off Modeling

The analysis and experimental results indicate that the IGBT turn-on time is primarily determined by its parasitic capacitance, stray inductance, internal resistance of the driver, and bus voltage. Building on this knowledge and integrating the analytical results from Section 2.2 with Equations (8), (9), and (12), the output voltage of the driver can be characterized as follows:(26)$$U_{r}=\left\{\begin{array}{c} \begin{array}{cc} \;U_{\infty }\cdot \left(1-e^{-\frac{t}{RC}}\right), & when\;t<t_{m}\; \end{array}\\ \begin{array}{cc} \;U_{\infty }\cdot \left(1-e^{-\frac{t}{RC}}\right)+U_{\infty }\cdot K_{L}\cdot e^{-\frac{(t-t_{m})\cdot L}{R}}, & when\;t\geq t_{m} \end{array} \end{array}\right.$$
(27)$$U_{f}=\left\{\begin{array}{c} \begin{array}{cc} U_{0}\cdot \left(-e^{-\frac{t}{RC}}\right) & when\;t\leq t_{m} \end{array}\\ \begin{array}{cc} U_{0}\cdot \left(-e^{-\frac{t}{RC}}\right)-U_{f}(t_{m})\cdot e^{-\frac{(t-t_{m})\cdot L}{R}} & when\;t>t_{m} \end{array} \end{array}\right.$$
where $U_{\infty }$ represents the driver output steady-state voltage (depending on the bus voltage), R represents the IGBT drive resistance, C represents the IGBT equivalent parasitic capacitance, L represents the IGBT internal equivalent stray inductance, $K_{L}$ represents the inductive gain, $t_{m}$ represents the time of the on–off process to enter the Miller platform, and $U_{0}$ represents the initial moment of the decline in the driver output voltage.

Linear fitting of the experimentally measured $U_{\infty }$ at different bus voltages yields the $U_{\infty }-U_{dc}$ curve shown in Figure 8, which can be expressed as follows:(28)$$U_{\infty }=0.84\times U_{dc}-0.5976\;\;(V)$$

Using the experimental data $U_{R}$ obtained in Section 3.2 and Section 3.3, a MOPSO algorithm is employed to fit the parameters in the models described by Equations (26) and (27). The RMSE of the parameters at varying bus voltages are compared, and the mean values of each parameter that yield the smallest RMSE are considered as the estimated values of the model parameters. The fitting results are illustrated in Figure 9, demonstrating that the established Equations (26) and (27) accurately characterize the actual switching behaviors at different bus voltages. The parameters derived from the fitting process are presented in the Table 1.

## 4. Driver Error Compensation Method Based on IGBT Switching Characteristics

A new error compensation method for drivers is proposed to calculate the command and actual voltage errors in real time by modeling and analyzing the driver inputs and outputs. This method facilitates compensation to mitigate driver errors.

### 4.1. Driver Input–Output Error Model

SVPWM achieves various command voltage levels by adjusting the duty cycle. As shown in Figure 3c, the actual voltage output from the driver is governed by the principle of volt–second balance, which states that the driver’s output corresponds to the average voltage over a single PWM cycle. When the duty cycle of the upper bridge conductor is set as δ, the actual voltages of the driver output are as follows:(29)$$\begin{array}{c} U_{act}=\frac{1}{T_{pwm}}(\int_{0}^{\delta T_{pwm}-T_{d}}U_{r}dt+\int_{0}^{T_{d}}U_{f}dt-\int_{0}^{\left(1-\delta \right)T_{pwm}-T_{d}}U_{r}dt-\int_{0\;}^{T_{d}}U_{f}dt)\;. \end{array}$$

The corresponding command voltage is
(30)$$U_{ref}=\delta \cdot \frac{1}{2}U_{dc}-\left(1-\delta \right)\cdot \frac{1}{2}U_{dc}=\left(2\delta -1\right)\cdot \frac{1}{2}U_{dc}.$$

By substituting Equations (28), (29), and (31) into Equation (30), it is possible to derive the input‒output voltage model of the driver.
(31)$$\begin{array}{c} U_{act}=\frac{1}{T_{pwm}}\left\{2\rho U_{\infty }T_{pwm}\right.\\ \;\;+{RCU}_{\infty }\left(1+e^{-\frac{T_{d}}{RC}}\right)\left[e^{-\frac{\left(\rho +0.5\right)T_{pwm}-T_{d}}{RC}}-e^{-\frac{\left(0.5-\rho \right)T_{pwm}-T_{d}}{RC}}\right]\\ +\left[-\frac{U_{\infty }K_{L}R}{L}\left(1-e^{-\frac{T_{d}}{RC}}\right)+\left(K_{L}-1\right)\frac{U_{\infty }R}{L}\left(e^{-\frac{T_{d}L}{R}}-e^{-\frac{t_{m}L}{R}}\right)\right]\\ \cdot \left.\left[e^{-\frac{\left(\rho +0.5\right)T_{pwm}-T_{d}-t_{m}}{R}L}-e^{-\frac{\left(0.5-\rho \right)T_{pwm}-T_{d}-t_{m}}{R}L}\right]\right\}. \end{array}$$
where $\rho =\frac{U_{ref}}{U_{dc}}$.

The errors in the output voltage of the driver are as follows:(32)$$∆v=U_{ref}-U_{act}.$$

Substituting the model parameters obtained from the fitting estimation, $1/RC=1.5\times 10^{6}(1/s)$, $K_{L}=0.094$, $L/R=1.8\times 10^{4}(1/s)$, $t_{m}=1.5\times 10^{-6}(s)$, the configured dead time $T_{d}=5\times 10^{-6}(s)$, and the carrier period $T_{pwm}=0.0001(s)$, into Equation (31) yields:(33)$$\begin{array}{cc} U_{act} & =\left(0.005746U_{dc}-0.003933\right)\cdot \left(e^{-\frac{150U_{ref}}{U_{dc}}-67.5}-e^{\frac{150U_{ref}}{U_{dc}}-67.5}\right)\\ & -\left(0.0172U_{dc}-0.0117\right)\cdot \left(e^{-\frac{1.8U_{ref}}{U_{dc}}-0.783}-e^{\frac{1.8U_{ref}}{U_{dc}}-0.783}\right)\\ & +\left(1.7228U_{ref}-\frac{{1.1792U}_{ref}}{U_{dc}}\right)\left(V\right). \end{array}$$

The plot of $\frac{U_{ref}}{U_{dc}}$ versus ∆v at different bus voltages is shown in Figure 10a, where the change in voltage error can be approximated as a linear change over the point (0, 0) when the bus voltage is constant, such that compensation
(34)$$∆v=a\cdot \frac{U_{ref}}{U_{dc}}.$$

The results of fitting the error slope a are shown in Table 2. a at different bus voltages is fitted as in Figure 10b.
(35)$$a=0.1238\cdot U_{dc}+0.59967.$$

### 4.2. Voltage Compensation Method

The inverse Clark transform is applied to the command voltage within the αβ coordinate system, converting the command voltage into a three-phase voltage.
(36)$$\left[\begin{array}{c} V_{a}\\ V_{b}\\ V_{c} \end{array}\right]=\left[\begin{array}{cc} 1 & 0\\ -\frac{1}{2} & \frac{1}{2\sqrt{3}}\\ -\frac{1}{2} & -\frac{1}{2\sqrt{3}} \end{array}\right]\left[\begin{array}{c} u_{\alpha }^{*}\\ u_{\beta }^{*} \end{array}\right].$$

The three-phase voltages are introduced into Equations (31) and (32) to obtain the three-phase voltage error. The three-phase voltage error is subsequently transformed into the αβ coordinate system through the Clark transformation.
(37)$$\left[\begin{array}{c} {∆u}_{\alpha }\\ {∆u}_{\beta } \end{array}\right]=\frac{2}{3}\left[\begin{array}{ccc} 1 & -\frac{1}{2} & -\frac{1}{2}\\ 0 & \frac{\sqrt{3}}{2} & -\frac{\sqrt{3}}{2} \end{array}\right]\left[\begin{array}{c} {∆V}_{a}\\ {∆V}_{b}\\ ∆V_{c} \end{array}\right].$$

The driver output error can be compensated by changing $u_{\alpha }$ and $u_{\beta }$ in the SVPWM control in real-time according to Equations (34)–(38).
(38)$$\left\{\begin{array}{c} u=u_{\alpha }^{*}+{∆u}_{\alpha }\\ u_{\beta }=u_{\beta }^{*}+{∆u}_{\beta } \end{array}\right.$$

A schematic diagram of voltage compensation is provided in Figure 11.

## 5. Simulation Results and Analysis

Using the above drive error compensation method for simulation, MATLAB R2023a Simulink was used to build the speed loop and the current loop simulation model, and the simulation results of the motor parameters are shown in Table 3 (to ensure the consistency of experimental parameters, the motor parameters utilized in the simulation were aligned with those employed in the subsequent experiments). The bus voltage was 30 V, the dead time was set to 5 μs, the driver model was analyzed according to the PM25RLA120 switching characteristics with the non-ideal characteristics of the driver, and the input was a three-phase sinusoidal voltage with an amplitude of 10 V.

As the ∆v of this compensation method adjusts in real time by varying the command voltage, the input is a three-phase sinusoidal voltage that simulates the change in command voltage. The effect of this method under different command voltages can be evaluated by comparing the input and output data before and after compensation.

Figure 12a shows a comparison of the A-phase voltages before and after compensation. Without compensation, the voltage error fluctuates, with a peak value of approximately 2 V, largely due to the non-ideal characteristics of the driver, such as internal voltage drop and circuit loss. After driver error compensation, the actual phase voltage closely aligns with the command voltage, reducing the voltage error to a peak value of approximately 0.3 V. Additionally, the RMSE between the actual voltage and the ideal phase voltage decreases from 0.7345 V to 0.0879 V. A similar error reduction is observed in the A-phase current before and after compensation, as shown in Figure 12b. This finding demonstrates that the compensation method effectively reduces the voltage loss and current error caused by the internal structure and switching characteristics of the IGBT.

The current loop was closed, and a sinusoidal command current with an amplitude of 0.05 A and a bias of 0.06 A was input to the q-axis. A comparison of the results before and after compensation, as demonstrated in Figure 13a, indicates a significant reduction in current error during the tracking of the command current. This finding highlights the effectiveness of the compensation method in enhancing current accuracy.

The speed loop was closed, with the command speed set to 0.1 r/s. The motor was operated in constant torque mode, maintaining a torque of 0.1 N/m. A comparative analysis of tracking speeds before and after compensation, as illustrated in Figure 13b, reveals a marked decrease in fluctuation error during both the rising and stabilizing phases of the speed following compensation relative to the errors observed prior to compensation. These results highlight the method’s efficacy in improving speed accuracy and stability.

This method effectively reduces both voltage loss and current error attributed to the internal structure and switching characteristics of the IGBT. Moreover, it enhances the overall performance and stability of the system.

## 6. Experimentation Results and Analysis

The PMSM drive control platform was developed using a DSP (TMS320F28335) as the controller. The experimental platform is depicted in Figure 14. The driver used is the PM25RLA120 (Manufactured by Mitsubishi, located in Tokyo, Japan), and the PMSM employed is the LD-170095 (Manufactured by ZCOE, located in Qingdao, China) torque motor, with the respective parameters detailed in Table 4.

In the experiment, the bus voltage was set at 20 V, the PWM input frequency was configured to 10 kHz, and the dead time was set to 5 μs. A magnetic powder damper was used to couple the PMSM, with the damper current established at 0.45 A for the speed loop experiment. Additionally, the speed waveform was sinusoidal, with an amplitude of 20 degrees and a frequency of 0.1 Hz.

As shown in Figure 15a, the commutation process of friction moment perturbation occurs when the speed is nearly zero, influenced by the friction present within the system. At this point, the direction of the friction force experiences a sudden change. The system’s inability to promptly track the current in response to the fluctuations in the friction force, due to the stringent constraints of robust predictive current control, leads to significant fluctuations. The speed loop speed RMSE demonstrates a notable decrease, from 0.0130°/s prior to compensation to 0.0061°/s following compensation.

To further substantiate the stability and universality of this method, the bus voltage was increased to 30 V, and the previously described experiment was repeated. As illustrated in Figure 15b, the speed loop following effect after compensation shows significant improvement compared to the state before compensation. However, there is a decrease in the compensation effect when compared to the results obtained at 20 V. This decline can be attributed to the fact that as the actual bus voltage increases, the ratio of the error amount to the commanded voltage decreases (as indicated by the error slope a in Equation (34)), thereby enhancing the control effect prior to compensation. Furthermore, the system is capable of maintaining minimal speed fluctuations across various operational conditions, thus improving the overall stability and performance of the system. Figure 15 illustrates that during the speed loop tracking process, speed fluctuations manifest at specific intervals, resulting in considerable errors (for instance, at t = 5 s and t = 6 s), which can be attributed to the influence of interference events during operation. An analysis of these interference events will be discussed in future publications.

Multiple experiments were conducted at different bus voltages, and the results were largely consistent, suggesting that only representative experimental results were presented.

To further investigate the reasons behind the reduction in speed fluctuation, the load was decreased during the experiments, allowing the motor to operate under relatively smooth speed variations. The modes of the composite voltage vector ($u_{s}=\sqrt{{u_{\alpha }}^{2}+{u_{\beta }}^{2}}$) were analyzed throughout the experimental process, as shown in Figure 16. This figure highlights that the composite voltage vector notably changes following compensation. First, the amplitude of high-frequency noise is significantly reduced, indicating that interference during motor operation has been effectively mitigated. Second, the fluctuation range of the output voltage is considerably diminished. This provides further evidence that the error in the voltage output has been effectively reduced after compensation, leading to improved accuracy of the voltage output and smoother control.

In conclusion, the comparative analysis of the experimental results before and after compensation demonstrates that the compensation method can significantly reduce the speed error of the speed loop while enhancing the control accuracy and stability of the system. Moreover, this method exhibits excellent adaptability and robustness, allowing it to maintain superior performance under diverse operational conditions. These advantages suggest that the compensation method holds considerable potential for practical applications.

## 7. Conclusions

This paper conducts an in-depth analysis of the primary factors influencing driver input–output errors and successfully constructs a mathematical model based on these factors. Addressing the non-ideal characteristics of PMSM drivers, an innovative error compensation method is proposed.

This method is capable of designing voltage compensation strategies tailored to different types of drivers. The use of a feed-forward voltage compensation technique effectively mitigates the negative effects of driver input–output errors on the control system.

Both the simulation and experimental results demonstrate that the proposed approach successfully alleviates motor operational issues arising from the non-ideal characteristics of the drive. Furthermore, this method can be implemented through software alone, eliminating the need for additional hardware, which enhances its practicality and broadens its potential applications.

This study proposes a novel concept for error compensation in PMSM drivers, marking a significant advancement in enhancing the performance and stability of motor control systems.

In conclusion, we will implement several enhancements to the experiments presented in this study. First, we will expand the range of experimental equipment utilized in application experiments, thereby enabling a more comprehensive evaluation of the universality of the compensation methods. Second, we plan to integrate this compensation method within a complete control system to facilitate more systematic and thorough experimentation. This approach will assist in validating the effectiveness and reliability of the compensation strategy in practical applications. Finally, considering the impact of disturbance torque on the experimental results, our future research will focus on estimating and compensating for disturbances, including disturbance torque, with the aim of minimizing their influence on the experimental outcomes. This strategy aims to further improve the control accuracy and robustness of permanent magnet synchronous motors.

## Figures and Tables

**Figure 1 sensors-24-07945-f001:**
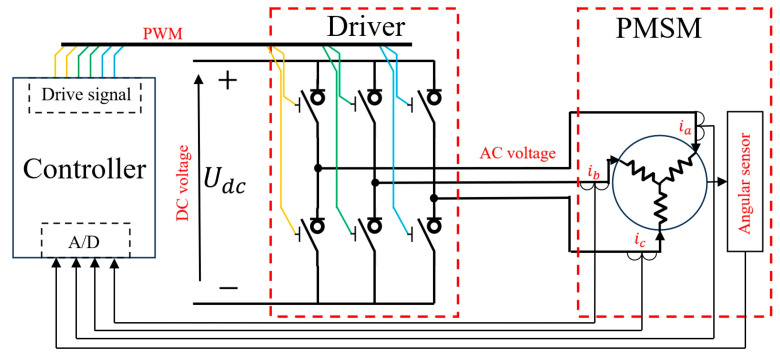
Drive–control system for PMSMs.

**Figure 2 sensors-24-07945-f002:**
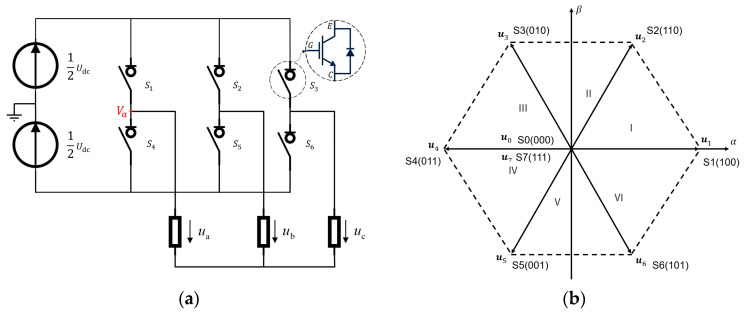
(**a**) Equivalent circuit diagram of the SVPWM driver. (**b**) Vector diagram of the space voltage.

**Figure 3 sensors-24-07945-f003:**
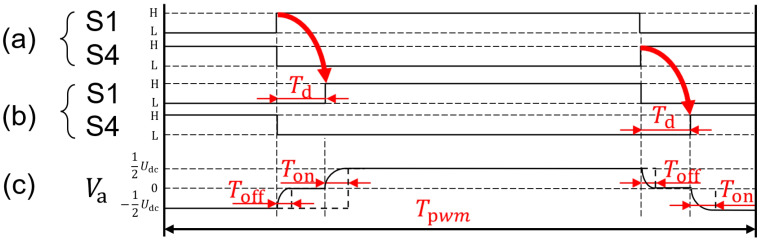
Dead time effect on the IGBT conduction and output voltages. (**a**) The ideal control signals for the pair of switches. (**b**) The control signals with dead time for the pair of switches. (**c**) Electric potential of the $S_{1}-S_{4}$ connection point (a phase terminal).

**Figure 4 sensors-24-07945-f004:**
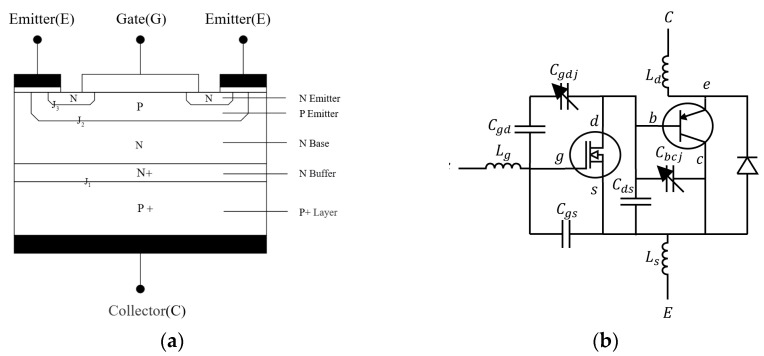
Schematic diagram of the internal structure of an IGBT. (**a**) Basic structure; (**b**) equivalent circuit.

**Figure 5 sensors-24-07945-f005:**
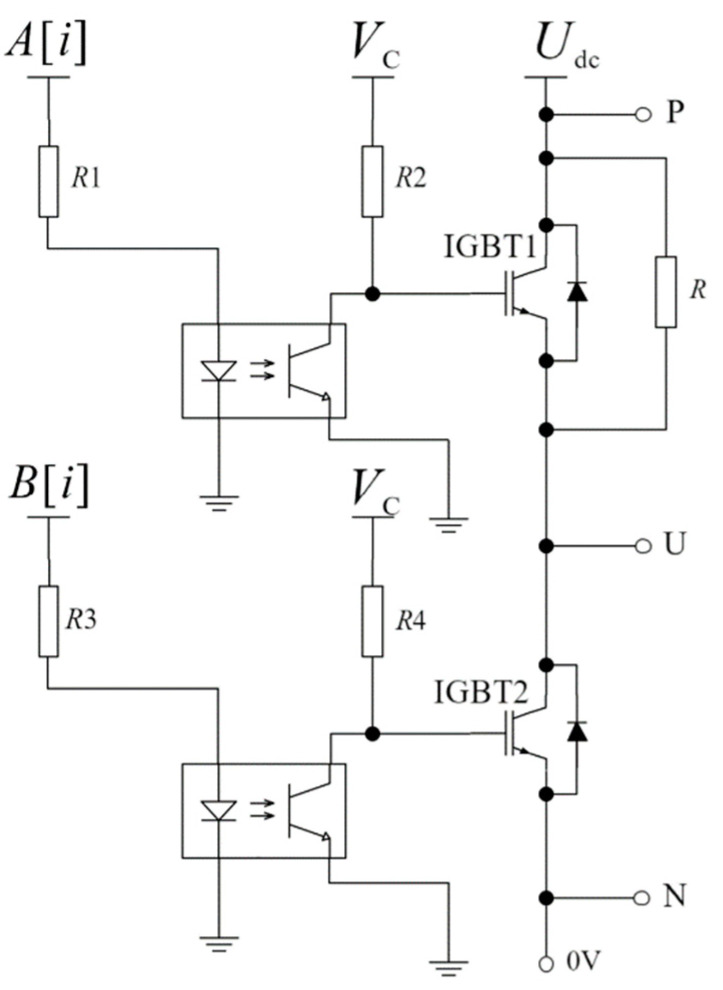
IGBT on–off time test circuit.

**Figure 6 sensors-24-07945-f006:**
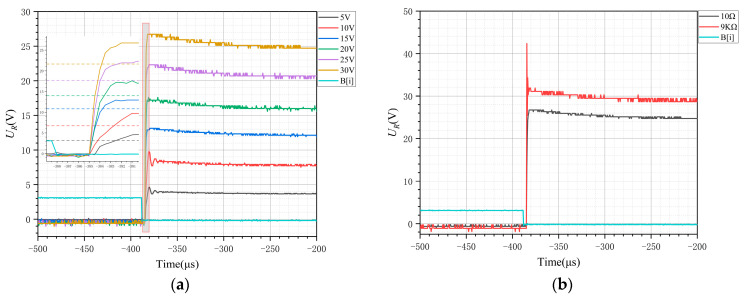
(**a**) Comparison of the voltage $U_{R}$ when switching on (R=10 Ω). (**b**) Switch-on voltages for different resistances ($U_{dc}=30\;V$).

**Figure 7 sensors-24-07945-f007:**
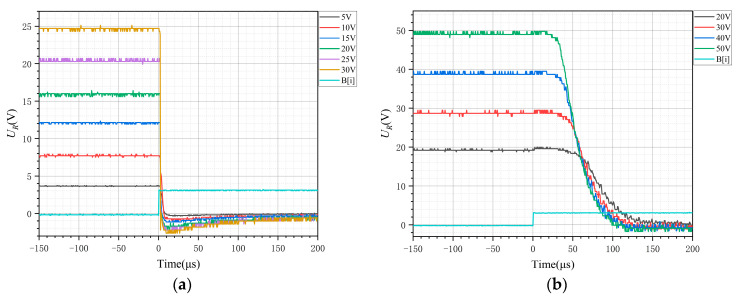
(**a**) Comparison of the voltage $U_{R}$ when switching off (R=10 Ω). (**b**) Comparison of the voltage $U_{R}$ when switching off (R=9 KΩ).

**Figure 8 sensors-24-07945-f008:**
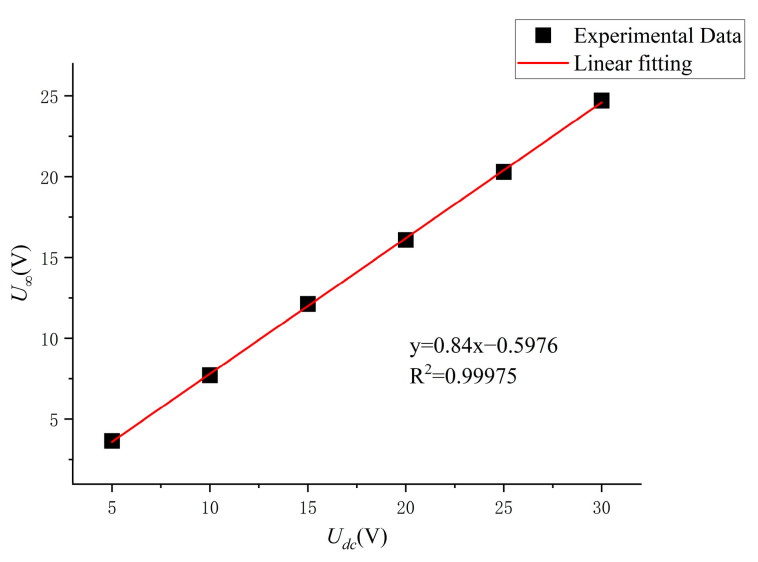
Steady-state voltage $U_{\infty }-U_{dc}$ relationship curve.

**Figure 9 sensors-24-07945-f009:**
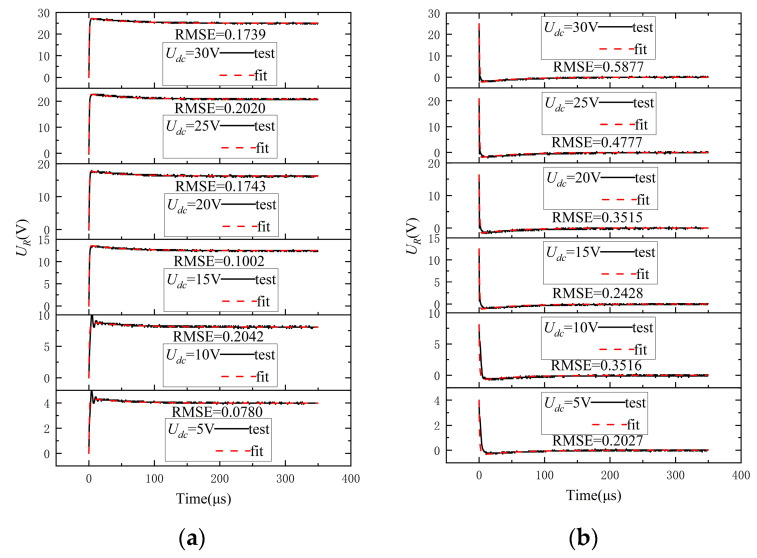
(**a**) Fitting results of the turn-on process. (**b**) Fitting results of the turn-off process.

**Figure 10 sensors-24-07945-f010:**
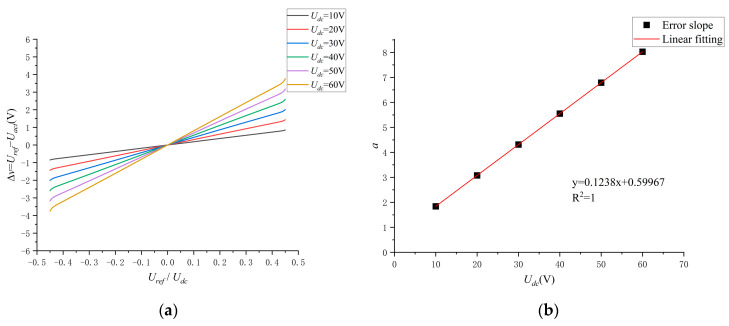
(**a**) Graph of the pattern of change in ∆v. (**b**) Fitting result (a).

**Figure 11 sensors-24-07945-f011:**
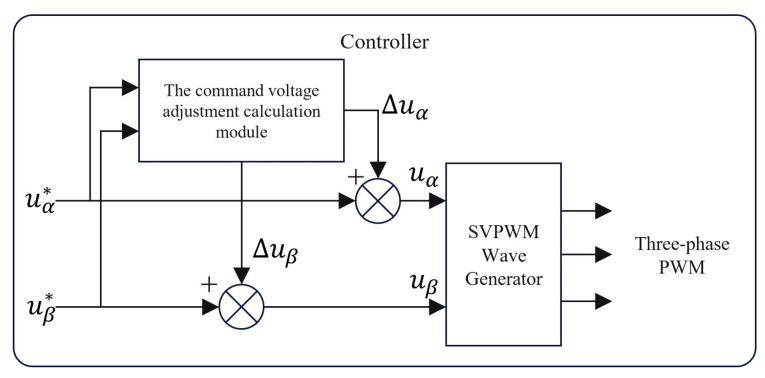
Schematic diagram of voltage compensation.

**Figure 12 sensors-24-07945-f012:**
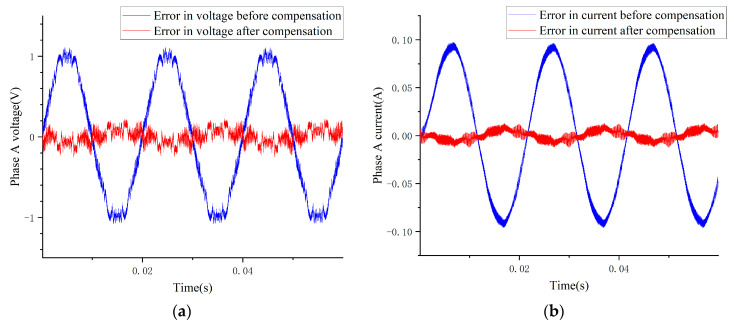
(**a**) Phase voltage error before/after driver compensation. (**b**) Phase current error before/after driver compensation.

**Figure 13 sensors-24-07945-f013:**
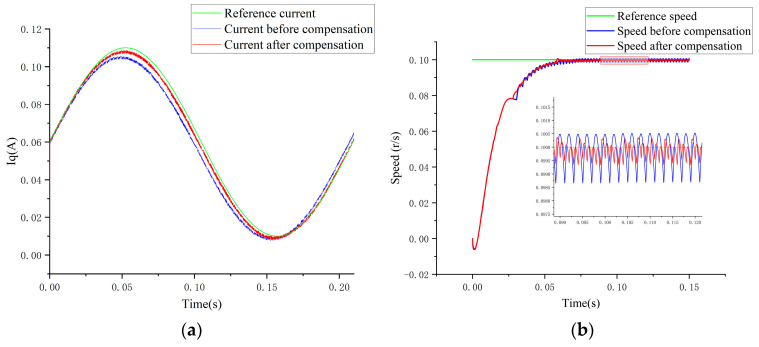
(**a**) q-axis current before/after driver compensation. (**b**) Speed before/after driver compensation.

**Figure 14 sensors-24-07945-f014:**
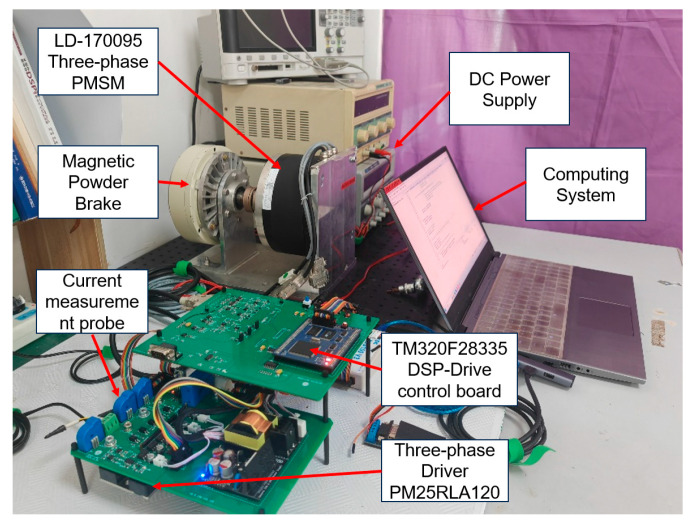
Experimental platform.

**Figure 15 sensors-24-07945-f015:**
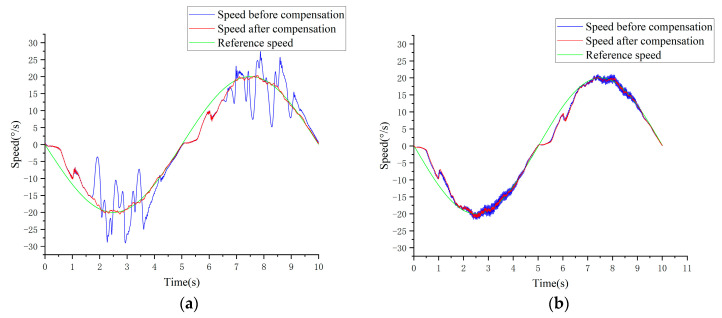
(**a**) Comparison of speed loop performance $U_{dc}=20\;V$. (**b**) Comparison of speed loop performance $U_{dc}=30\;V$.

**Figure 16 sensors-24-07945-f016:**
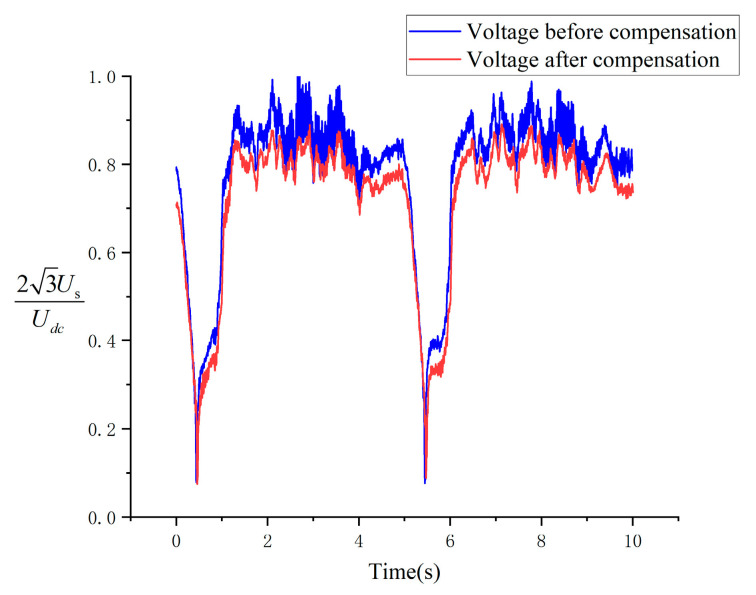
Magnitude of the synthetic voltage vector.

**Table 1 sensors-24-07945-t001:** The fitting parameters in Equations (26) and (27).

Parameters	Value
$K_{L}$	0.09
R·C	$6.67\times 10^{-7}\Omega \cdot F$
$\frac{L}{R}$	$5.56\times 10^{-5}\;H/\Omega$
$t_{m}$	$2.98\times 10^{-7}\;s$

**Table 2 sensors-24-07945-t002:** Linear fitting results for voltage error at each busbar.

$U_{dc}(V)$	a	$R^{2}$
10	1.83765	0.99998
20	3.07563	0.99997
30	4.31361	0.99996
40	5.55159	0.99996
50	6.78957	0.99996
60	8.02755	0.99996

**Table 3 sensors-24-07945-t003:** Motor parameters in the simulation.

Name	Value
Stator Equivalent Resistance	9.9 Ω
Stator Equivalent Inductance	17.9 mH
Continuous Torque	23.8 N·m
Torque Constant	9.5 N·m/A
Moment of Inertia	0.02 kg∙m^2^
Number of Pole Pairs	15
Viscous Damping Coefficient	0.001

**Table 4 sensors-24-07945-t004:** Experimental platform.

Module	Name	Value
LD-170095	Stator Equivalent Resistance	9.9 Ω
	Stator Equivalent Inductance	17.9 mH
	Continuous Torque	23.8 N·m
	Continuous Current	2.5 A
	Maximum Torque	71 N·m
	Maximum Current	7.5 A
	Torque Constant	9.5 N·m/A
	Moment of Inertia	0.02 kg∙m^2^
	Number of Pole Pairs	15
	Maximum Rated Power Consumption	30.1 W
	Maximum Rotational Speed	190 rpm
PM25RLA120	Maximum Collector-Emitter Voltage	1200 V
	Collector Current	15 A
	Collector Current (Peak)	30 A
	Recommendation dead time	≥2.5 μs
	Recommendation PWM input frequency	≤20 kHz

## Data Availability

Data are contained within this article.

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
