# Peer review of "Characteristic Analysis and Error Compensation Method of Space Vector Pulse Width Modulation-Based Driver for Permanent Magnet Synchronous Motors"

_sensors, 2024, doi:10.3390/s24247945_

Round 1

Reviewer 1 Report

Comments and Suggestions for Authors

This paper describes an analysis and error compensation method of a space vector pulse width modulation-based driver for a permanent magnet synchronous motor. I suggest to improve the paper based on the major modifications listed above:

1.       In line 316, the authors wrote that a 9,9 Ω value can approximate the motor resistance. It is necessary to explain why this value was selected and the data used for this approximation.

2.       The authors refer to their method as feedforward, while Figure 11 shows that the correction is added to the signal and does not bypass any dynamics.

3.       The driver applied in this paper is PM25RLA120. During datasheet analysis, it can be seen that it can operate up to 1200 V. On the other hand, experimental tests shown in the paper are conducted for 30 VDC. Therefore, the potential of the driver is not fully utilized. Finally, different modules should be used for low-voltage applications with much lower dead times and losses.

4.       The shape of the q-axis current in Figure 13 (a) indicates the valid operation of the PMSM drive.

5.       In the speed waveform shown in Figure 13 (b), oscillations are visible, e.g., around t = 0 s, t = 0.03 s, and t = 0.05 s.

6.       In Figure 15, the speed unit is incorrect. How can the authors explain the sudden speed fluctuations after compensation visible at t = 1 s, t = 5 s, and t = 6 s?

Reviewer 2 Report

Comments and Suggestions for Authors

The objective of this paper is to develop space vector pulse width modulation-based driver.So this work conducted theoretical analysis and experiments on the error compensation method.

This paper is well structured, and their research is interesting and has significance.

I would recommend publishing this manuscript only after its improvement by following the suggestions described below:

1. The practical value of this research work should be clarified and highlighted in the Abstract, which can help readers understand the engineering background of this research work.

2. How did authors choose these motor parameters in the simulatio? Please clarify whether these parameters are representative.

3.  For better transparency, all values used in equations could also be presented in a table.

4. In section 6, the authors can supply much more detail on test device.

5. In section 7, please clarify some improvement to the experiment in the future.

6. In its current state, the level of English throughout this manuscript should be checked for grammar, style and syntax.

Round 2

Reviewer 1 Report

Comments and Suggestions for Authors

All of my comments have been addressed, and the paper may be accepted for publication in its current form. Congratulations.